# Photoactivatable BODIPYs for Live-Cell PALM

**DOI:** 10.3390/molecules28062447

**Published:** 2023-03-07

**Authors:** Yang Zhang, Yeting Zheng, Andrea Tomassini, Ambarish Kumar Singh, Françisco M. Raymo

**Affiliations:** 1Program of Polymer and Color Chemistry, Department of Textile Engineering, Chemistry and Science, North Carolina State University, Raleigh, NC 27606, USA; 2Laboratory for Molecular Photonics, Department of Chemistry, University of Miami, 1301 Memorial Drive, Coral Gables, FL 33146-0431, USA

**Keywords:** borondipyrromethenes (BODIPYs), fluorescence imaging, photoactivatable fluorophores (PAFs), photoactivated localization microscopy (PALM), single-molecule localization microscopy (SMLM)

## Abstract

Photoactivated localization microscopy (PALM) relies on fluorescence photoactivation and single-molecule localization to overcome optical diffraction and reconstruct images of biological samples with spatial resolution at the nanoscale. The implementation of this subdiffraction imaging method, however, requires fluorescent probes with photochemical and photophysical properties specifically engineered to enable the localization of single photoactivated molecules with nanometer precision. The synthetic versatility and outstanding photophysical properties of the borondipyrromethene (BODIPY) chromophore are ideally suited to satisfy these stringent requirements. Specifically, synthetic manipulations of the BODIPY scaffold can be invoked to install photolabile functional groups and photoactivate fluorescence under photochemical control. Additionally, targeting ligands can be incorporated in the resulting photoactivatable fluorophores (PAFs) to label selected subcellular components in live cells. Indeed, photoactivatable BODIPYs have already allowed the sub-diffraction imaging of diverse cellular substructures in live cells using PALM and can evolve into invaluable analytical probes for bioimaging applications.

## 1. Introduction

The fluorescence microscope has become an indispensable tool in the biomedical laboratory to probe the dynamics and structures of biological samples at the micrometer level [1]. Indeed, the minimally invasive character, fast response and high sensitivity of fluorescence measurements [2], in conjunction with the availability of vast libraries of fluorescent probes and labeling kits from commercial sources [3], have turned fluorescence microscopy into the analytical method of choice to unravel the fundamentals of cellular processes [4]. Nonetheless, the spatial resolution of conventional fluorescence imaging schemes is, at best, two orders of magnitude larger than the dimensions of molecules. As a result, the structural factors regulating cellular functions at the molecular level remain elusive.

Optical diffraction [5] is the fundamental physical phenomenon that prevents the spatial resolution of fluorescence microscopes, based on far-field optics, from being significantly smaller than the wavelength of the emitted light [1]. Specifically, the objective lens of the microscope projects exciting radiation on the sample and collects the resulting emitted light (Figure 1a). The latter is focused in the form of a diffraction pattern (Figure 1b), called the Airy Pattern, with physical dimensions that are predominantly controlled by the wavelength of the emitted radiation. The radius of the inner disk of the Airy Pattern is approximately half of the emitted wavelength, which happens to be in the visible region of the electromagnetic spectrum for most bioimaging applications. It follows that the Airy pattern of a single fluorescent molecule is hundreds of nanometers in size. As a result, distinct fluorescent molecules can be resolved only if their distance is greater than the sum of the radii of their Airy disks.

The advent of super-resolution imaging [6,7,8] provided viable strategies to overcome diffraction and visualize structural features at the nanoscale with transformative implications in biology and medicine. Indeed, the resulting imaging methods allow the acquisition of fluorescence images with sub-diffraction resolution by differentiating closely spaced fluorophores in time. Some of these methods are based on the localization of single molecules with switchable fluorescence to reconstruct subdiffraction images and are collectively called single-molecule localization microscopy (SMLM) [9]. One of these SMLM strategies relies on photochemical reactions to switch the fluorescence of single molecules and is termed photoactivated localization microscopy (PALM) [10]. This ingenious and powerful methodology, however, cannot be implemented with conventional fluorophores. It demands, instead, biocompatible emissive probes with photochemical and photophysical properties specifically engineered for this application [11]. The stringent requirements for the implementation of PALM can be satisfied with appropriate structural modifications of the basic scaffold of the borondipyrromethene (BODIPY) chromophore [12,13,14,15,16,17,18,19]. In this review, we illustrate the structural designs and operating principles of all BODIPY derivatives with photoactivatable fluorescence developed so far for PALM of live cells.

## 2. Photoactivated Localization Microscopy

PALM overcomes diffraction relying on fluorescence photoactivation and single-molecule detection to enable the visualization of biological samples with spatial resolution at the nanometer level [6,9,10,11]. Specifically, a given sample of interest is labeled with probes capable of switching from a nonemissive state to an emissive form under irradiation at an appropriate activation wavelength (λ_Ac_). The entire field of view is then illuminated at λ_Ac_ with sufficiently-low power densities to activate only a sparse subset of probes (Figure 2a,b) and ensure negligible probability of finding emissive species at subdiffraction separations. If brightness and photon budget of the emissive species are sufficiently high for single-molecule detection and localization with nanometer precision, respectively, irradiation at their excitation wavelength (λ_Ex_) then allows the determination of their spatial coordinates (Figure 2b,c). Further excitation eventually bleaches the localized probes and permanently turns off their fluorescence (Figure 2c,d). A new subset of probes is then activated, localized and bleached. The very same sequence of three steps is reiterated thousands of times to provide multiple subsets of localized coordinates (Figure 2e–h), which can ultimately be combined into a reconstructed subdiffraction image (Figure 2i).

The overall result of PALM is the ability to visualize structural features with a spatial resolution that would be impossible to achieve with conventional fluorescence imaging schemes [6,9,10,11]. The downside of this transformative strategy, however, is that the probes required for its implementation must satisfy stringent photochemical and photophysical requirements. Indeed, they must switch from nonemissive to emissive states under illumination conditions that, ideally, do not cause any photodamage to biological samples. Furthermore, the emissive form must have a large molar absorption coefficient (ε) at λ_Ex_ and a high fluorescence quantum yield (ϕ_Fl_) for the brightness (ε × ϕ_Fl_) to be appropriate for single-molecule detection. Additionally, the emissive species must also be able to tolerate multiple excitation/deactivation cycles to allow the detection of hundreds of emitted photons per molecule and, hence, permit localization with nanometer precision. Engineering such a unique combination of properties into a single molecular construct, while retaining compatibility with biological environments and labeling technologies, is a daunting task.

## 3. Photoactivatable Fluorophores

Photoactivatable fluorophores (PAFs) switch from a nonemissive to an emissive state under illumination at an appropriate activation wavelength (λ_Ac_ in Figure 3) [20,21,22,23,24,25,26,27,28,29,30,31,32,33,34,35,36]. The latter species produces fluorescence upon irradiation at the corresponding excitation wavelength (λ_Ex_ in Figure 3). As a result, PAFs enable the photoactivation of fluorescence in a defined region of space at a precise interval of time, relying on the interplay of lasers operating at λ_Ac_ and λ_Ex_. In turn, the spatiotemporal control of fluorescence permits the implementation of imaging strategies that would be impossible to perform with conventional fluorophores. Indeed, PAFs allow the sequence of steps required to (1) overcome diffraction in PALM (Figure 2) [10], (2) track dynamic events in fluorescence photoactivation and dissipation (FPD) [37], (3) monitor fluid flow in microscaled channels in flow-tagging velocimetry [38] and (4) write and read optical barcodes in living organisms [39].

PAFs for irreversible fluorescence photoactivation (Figure 3a) are generally constructed by integrating a fluorescent chromophore (fluorophore in Figure 3) and a photocleavable protecting group (photocage in Figure 3) within the same molecular skeleton [24,27]. Upon illumination at λ_Ac_, the photocage separates from the fluorophore, enabling the latter to produce fluorescence upon irradiation at λ_Ex_. PAFs for reversible fluorescence photoactivation (Figure 3b) are instead assembled by covalently connecting a fluorophore to a photochromic chromophore (photochrome in Figure 3). The latter component interconverts reversibly between two distinct states upon illumination at λ_Ac_, allowing the fluorophore to emit under irradiation at λ_Ex_ only in one of the two interconvertible states.

In both structural designs for fluorescence photoactivation, intramolecular quenching can be exploited to encourage the nonradiative deactivation of the fluorophore and prevent fluorescence in the initial state of the PAF. The physical separation of the photocage from the fluorophore (Figure 3a) or the conversion of the photochrome (Figure 3b) suppresses the quenching pathway, permits the radiative deactivation of the fluorophore and generates fluorescence. Based on this mechanism, the absorption (Figure 4a,c) of the fluorophore remains essentially unaffected with photoactivation, while the emission (Figure 4b,d) increases significantly. Alternatively, the photoinduced disconnection of the photocage or conversion of the photochrome can be engineered to shift bathochromically the absorption of the fluorophore (Figure 4e,g). Irradiation at a λ_Ex_ within the shifted absorption band selectively excites the photochemical product to switch fluorescence from off to on (Figure 4f,h). Indeed, the lack of any absorbance at λ_Ex_ for the initial state of the PAF eliminates any background fluorescence, that is unavoidable (Figure 4b) for systems based on intramolecular quenching. In fact, unitary quenching efficiencies are hardly achievable, limiting the contrast levels accessible with photoactivation mechanisms based on intramolecular quenching. In turn, a limited contrast would complicate the single-molecule detection of a few strongly-emissive probes in the presence of a large excess of weakly-emissive species, which is an essential step (Figure 2b,c) of PALM [10]. Additionally, the lack of any significant change in absorption with photoactivation (Figure 4a,c) prevents the selective bleaching of the photoactivated species under irradiation at λ_Ex_, which is also an essential step (Figure 2c,d) of PALM. Thus, PAFs, operating on the basis photoinduced bathochromic absorption shifts rather than intramolecular quenching, should ideally be the photoactivatable probes of choice for PALM. Nonetheless, both photoactivation mechanisms have been used so far to reconstruct PALM images of live cells with photoactivatable cyanine [40,41], dihydrofuran [42], oxime [43], rhodamine [44,45,46,47,48], thioimide [49] and BODIPY [50,51,52,53] derivatives. The latter family of PAFs is the focus of this review.

## 4. Borondipyrromethenes

BODIPY chromophores consist of two pyrrole heterocycles bridged by a pair of carbon and boron atoms [12,13,14,15,16,17,18,19]. Symmetrical BODIPYs are generally synthesized from an aldehyde or acyl chloride precursor and two equivalents of the corresponding pyrrole heterocycle (Figure 5). Unsymmetrical BODIPYs can similarly be prepared from two different pyrrole precursors, as long as one of the two incorporates an acyl group in the α-position relative to the nitrogen atom. After condensation of the three or two molecular fragments into a single covalent skeleton, the boron atom must be introduced to lock the two heterocyclic subunits in a co-planar arrangement and allow electronic delocalization across the entire molecular construct. The resulting chromophoric platforms absorb in the green region of the electromagnetic spectrum with molar absorption coefficient (ε) close to 70,000 M^™1^ cm^™1^, when the substituents (R^1^–R^3^) of the pyrrole heterocycle do not allow further electronic delocalization. When extended electronic conjugation is instead possible over R^1^, R^2^ and/or R^3^, the absorption bands of BODIPYs shift to the red and, in some instances, can even reach the near-infrared region with ε approaching 100,000 M^™1^ cm^™1^.

The two ligands (R^4^) on the boron center are generally fluorine atoms, albeit numerous derivatives with alkoxy, alkyl or aryl groups on this position have been reported already [12,13,14,15,16,17,18,19]. The associated pair of (B–N) bonds, holding the boron atom in place and maintaining the co-planar arrangement, tolerate a wide range of experimental conditions. As long as strong acids or strong nucleophiles are avoided, the chromophoric platform generally remains intact and, therefore, can be modified with a wealth of synthetic procedures, providing access to an essentially infinite library of BODIPY derivatives.

The group (R^5^) on the *meso*-position of the BODIPY is generally a hydrogen atom, an alkyl substituent or an aryl ring [12,13,14,15,16,17,18,19]. In the latter instance, the conformational freedom about the dihedral angle between polycyclic chromophore and aryl substituent provides nonradiative pathways for the relaxation of the BODIPY first singlet excited state (S_1_). Steric hinderance, often in the form of methyl groups at R^1^, prevents these processes and ensures efficient radiative deactivation instead, as long as heavy atoms and/or electron acceptors/donors are not incorporated in any of the other substituents (R^2^–R^5^). The former atoms promote intersystem crossing from S_1_ to the first triplet excited state (T_1_). The latter groups can transfer electrons to/from S_1_. Both processes result in efficient fluorescence quenching. When the structural requirements to suppress processes competing with fluorescence are satisfied, BODIPY chromophores emit with high ϕ_Fl_, which can often approach unity. In turn, the large ε and high ϕ_Fl_ translate into appropriate brightness (ε × ϕ_Fl_) levels for single-molecule detection.

The fluorescence of BODIPY fluorophores can be photoactivated with the aid of compatible photocages (Figure 2a) relying on either intramolecular quenching (Figure 3a) or bathochromic shifts (Figure 3b) [35]. Indeed, several photoactivatable BODIPYs have already been designed around these operating principles and reported in the literature [39,50,51,52,53,54,55,56,57,58,59,60,61,62,63,64]. We implemented both photoactivation mechanisms, relying on the established photochemistry of *ortho*-nitrobenzyl (ONB) photocages [65]. For example, compound **1** (Figure 6) incorporates a pyridinium ring on the *meso*-position of a BODIPY chromophore and strongly absorbs green light (Figure 4a) [55]. Upon irradiation at a λ_Ex_ of 470 nm, electron transfer from the excited BODIPY to the adjacent pyridinium cation occurs efficiently. As a result, **1** emits with a ^In^ϕ_Fl_ of only 0.05 (Table 1). Upon illumination at λ_Ac_ in the 300–410 nm range, the ONB group connected to the nitrogen atom of the pyridinium quencher cleaves irreversibly to generate **2** and **3**. The photoinduced conversion of the pyridinium cation of **1** into the pyridine ring of **2** suppresses the quenching pathway. In fact, the latter compound emits with a ^Fi^ϕ_Fl_ of 0.50 (Table 1), corresponding to a contrast (^Fi^ϕ_Fl_/^In^ϕ_Fl_) of 100. It follows that the photolytic transformation of **1** into **2** is accompanied by a gradual increase in emission intensity (Figure 4b–g).

The photoinduced conversion of **1** into **2** causes the emission band of the BODIPY fluorophores to increase from low to high intensity (Figure 6b,g) with a contrast of (^Fi^ϕ_Fl_/^In^ϕ_Fl_) of 100 [55]. The photochemical transformation of **4** into **5** causes instead the emission band of the BODIPY fluorophore to bathochromically shift (Figure 7c,d) [57]. As a result, fluorescence switches from off to on (i.e., infinite contrast) in the spectral region where the only the photochemical product emits. Compound **4** also incorporates an ONB photocage within its molecular skeleton. Under irradiation at λ_Ac_ in the 300–410 nm range, the ONB groups cleaves irreversibly to generate **5** and **6**. This structural transformation converts the chiral *sp*^3^ carbon atom at the junction of the indole and oxazine heterocycles in **5** to *sp*^2^ in **6** and, only then, allows the BODIPY chromophore to extend electronic delocalization over the adjacent indole heterocycle. It follows that the photoinduced conversion of **4** into **5** bathochromically shifts the absorption and emission bands of the BODIPY fluorophore (Figure 7a–d). In fact, the reactant and product of this photochemical reaction emit with ^In^ϕ_Fl_ and ^Fi^ϕ_Fl_ of 0.07 and 0.50 (Table 1), respectively, upon irradiation at a λ_Ex_ of 480 nm.

## 5. Live-Cell Imaging

Compound **7** (Figure 8) was the first BODIPY derivative to have been photoactivated in live cells [54]. This molecule incorporates two hydrophilic 2-carboxyethyl substituents to ensure sufficient aqueous solubility for administration into the extracellular matrix as well as to avoid nonspecific adsorption on the many hydrophobic components in the intracellular environment. It also has a ligand for the selective covalent labeling of SNAP-tag proteins and a photocleavable ONB group. Incubation of live HeLa cells, expressing epidermal growth factor receptor (EGFR)–SNAP-tag fusions, with compound **7** results in the covalent attachment of the photoactivatable BODIPY to the protein targets, after the nucleophilic displacement of the guanine leaving group. A confocal laser scanning microscopy (CLSM) image (Figure 8a) of the labeled cells, recorded after extensive washing of unreacted probes out of the cells, reveals negligible fluorescence. The pair of electron-withdrawing nitro groups, engineered into the ONB photocage, promote electron transfer from the excited BODIPY and ensure low emission intensity. Control measurements on a model compound, without the SNAP-tag ligand, in phosphate buffer saline (PBS) suggest ^In^ϕ_Fl_ to be only 0.001 (Table 1). Upon illumination of the entire field of view at a λ_Ac_ of 365 nm, the ONB group separates from the BODIPY chromophore, suppressing the quenching pathway and increasing the emission intensity. The corresponding CLSM image (Figure 8b) clearly shows significant intracellular fluorescence. Consistently, the photoactivation of the model analogue results in a ^Fi^ϕ_Fl_ of 0.66 (Table 1), corresponding to a contrast (^Fi^ϕ_Fl_/^In^ϕ_Fl_) of 660.

The transformative experiments with **7** demonstrated that photoactivatable BODIPYs are, indeed, optimal probes for the spatiotemporal control of fluorescence in the interior of live cells [54]. However, only diffraction-limited fluorescence images were reported in these studies. The first examples of PALM images with photoactivatable BODIPYs were instead recorded with compound **11** (Figure 9) [50]. This molecule incorporates a paclitaxel ligand, capable of binding noncovalently the microtubules of live cells, as well as two photolabile ethyl groups on the boron atom of the BODIPY chromophore. These two groups promote the nonradiative deactivation of the excited fluorophore, presumably as a result of their conformational freedom. Control measurements in methanol suggest ^In^ϕ_Fl_ to be 0.002 (Table 1). Upon irradiation at a λ_Ac_ of 488 nm, the two ethyl groups are displaced by a pair of methoxy groups in methanol or hydroxy groups in water and, presumably, converted into ethane molecules. Control measurements in the former solvent indicate ^Fi^ϕ_Fl_ to be 0.96 (Table 1), corresponding to a contrast (^Fi^ϕ_Fl_/^In^ϕ_Fl_) of 480. The photoactivated fluorescence of this PAF could be exploited to capture diffraction-limited (Figure 9a) and reconstruct PALM (Figure 9b) images of the microtubules of live HeLa cells. Comparison of the two clearly reveals the drastic improvement in spatial resolution possible with PALM. Indeed, the full width at half maximum of the emission-intensity profiles (blue and red bars in Figure 9), measured across a microtubule in the two images, decreases from 453 to only 94 nm.

Compound **14** (Figure 10) is another remarkable example of photoactivatable BODIPY successfully employed to visualize intracellular targets with subdiffraction resolution in live cells [52]. This molecule has a photolabile tetrazine appendage connected to the *meso*-position of a BODIPY chromophore though a *para*-phenylene spacer. It also incorporates a ligand for the selective covalent labeling of HaloTag proteins. Control measurements in acetonitrile indicate ^In^ϕ_Fl_ to be only 0.002 (Table 1). Presumably, electron transfer from the tetrazine ring to the excited BODIPY chromophore is responsible for the predominant nonradiative deactivation of the chromophoric component. Upon illumination at a λ_Ac_ of 405 nm, however, the tetrazine appendage cleaves to release one molecule of acetonitrile and one of nitrogen. In turn, the photoinduced removal of the quencher causes a significant increase in emission intensity with a ^Fi^ϕ_Fl_ of 0.28 (Table 1), corresponding to a contrast (^Fi^ϕ_Fl_/^In^ϕ_Fl_) of 141. Incubation of live COS–K1 cells, expressing HaloTag on histone 2B (H2B), with compound **14** results in the covalent attachment of PAFs to the H2B–HaloTag fusions. After extensive washing of unreacted probes out of the labeled cells, irradiation at λ_Ac_ disconnects the tetrazine quenchers to enhance significantly the emission intensity of the BODIPY component. In turn, the photoactivated fluorescence can be exploited to acquire diffraction-limited (Figure 10a) and reconstruct PALM (Figure 10b) images of the sample. Comparing the two imaging modalities shows, once again, the significant improvement in spatial resolution possible through the sequence of photoactivation, localization and bleaching steps (Figure 2a–d) characteristic of PALM.

The fluorescence photoactivation mechanism of **7**, **11** and **14** relies on the suppression of a quenching pathway with the photoinduced cleavage of an ONB photocage, a pair of ethyl ligands or a tetrazine ring [50,52,54]. The inability to achieve unitary quenching efficiencies in the initial state, however, results in nonzero ^In^ϕ_Fl_. As a result, the photochemical products must be detected against some level of background fluorescence coming from the reactants. Additionally, these photochemical reactions have negligible influence on the spectral position of the BODIPY absorption, preventing the selective bleaching of the photoactivated species. Such drawbacks complicate the implementation of the sequence of steps (Figure 2a–i) required for PALM. Our mechanism for fluorescence photoactivation (Figure 7), based on a photoinduced bathochromic shift, overcomes both limitations [57]. In particular, we designed an analogue of our original photoactivatable BODIPY, in the shape of compound **17** (Figure 11), specifically for live-cell PALM [51]. This molecule has a styryl substituent, in place of the methyl group of the original molecule (cf., **4**), to position the absorption and emission bands of the photoactivated BODIPY in the red region of the electromagnetic spectrum, as well as to enhance its ε and facilitate single-molecule detection. Additionally, **17** incorporates also an *N*-hydroxysuccinimide ester (NHS) in position 5 of the indole heterocycle to enable the covalent connection of this compound to the primary amino groups of intracellular proteins. Indeed, incubation of live COS–7 cells with **17** results in the nonselective labeling of intracellular proteins and localization of the resulting conjugates in the lysosomal compartments. The characteristic fluorescence of the BODIPY chromophore in the spectral region where the initial state emits can clearly be observed, prior to photoactivation, in a diffraction-limited image (Figure 11a) of a single cell. The individual labeled organelles, however, have subdiffraction dimensions and cannot be resolved. The corresponding PALM counterpart (Figure 11b) shows instead single lysosomes with an average diameter of ca. 80 nm. This particular image was reconstructed from 100,000 subsets of coordinates acquired sequentially (Figure 2a–i) with a precision of ca. 15 nm in the single-molecule localization of the photoactivated probes.

The photocleavable groups of **1**, **4**, **7**, **14** and **17** require ultraviolet radiation or, at best, violet light for photoactivation [51,52,54,55,57]. Illumination of live cells with λ_Ac_ in this spectral range (350–410 nm), however, causes significant photodamage [66]. Compound **11** is a remarkable exception [50]. It can be photoactivated under illumination at a λ_Ac_ of 488 nm. Nonetheless, systematic cytotoxicity studies show that negligible cell mortality is observed only under irradiation at wavelengths longer than 500 nm with the typical power densities required for photoactivation [66]. Compound **20** (Figure 12) satisfies this crucial requirement [64]. It absorbs and emits in the red region of the electromagnetic spectrum with a ^In^ϕ_Fl_ of 0.90 (Table 1). Upon illumination at a λ_Ac_ of 630 nm, one of the two styryl substituents on the pyrrole heterocycles cleaves to generate **21**. The decrease in electronic delocalization with the photochemical conversion of **20** into **21** and **23** causes hypsochromic shifts in absorption and emission, albeit ^Fi^ϕ_Fl_ was not reported. In the interior of live cells, the photochemical product reacts further with intracellular thiols, such as glutathione, to the corresponding adducts (e.g., **23**), causing a further hypsochromic shift in absorption and emission. Indeed, CLSM images of live HeLa cells, recorded before (Figure 12a) and after (Figure 12b) photoactivation with λ_Ex_ of 561 and 488 nm, respectively, reveal fluorescence in both instances.

The ability to photoactivate fluorescence with red light is a significant advancement in the development of PAFs for bioimaging applications. However, the photoinduced hypsochromic shifts in absorption and emission, engineered into **20**, complicate the selective excitation of the photochemical product [64]. In turn, this drawback would complicate the single-molecule localization of the photoactivated species and their selective bleaching. In fact, only diffraction-limited images were reported in these seminal studies. The photochemical and photophysical properties of compound **24** (Figure 13) overcome some of these limitations; however, this PAF requires green, rather than red, light for photoactivation [53]. This molecule incorporates two BODIPY components connected through a sulfinyl group. It also has two aliphatic arms with terminal morpholine rings to encourage solubilization in aqueous environments and direct the overall molecular construct into the lysosomal compartments of live cells. The pair of BODIPY chromophores absorb and emit in the green region of the electromagnetic spectrum with a ^In^ϕ_Fl_ 0.20 (Table 1). Upon irradiation at a λ_Ac_ of 488 or 532 nm, the sulfinyl group is expelled to allow the direct connection of the two BODIPY fragments into a single chromophoric platform. As a result, the absorption and emission bands bathochromically shift from the green to the red region with a ^Fi^ϕ_Fl_ of 0.23 (Table 1). Incubation of live RAW264.7 cells with **24** results in the lysosomal localization of the PAFs. Illumination of the labeled cells at a λ_Ac_ of 488 nm can then be exploited to photoactivate the internalized probes and implement the sequence of steps (Figure 2a–i) required for the reconstruction of PALM images. Comparison of diffraction-limited (Figure 13a) and PALM (Figure 13b) images of the labeled cells shows, yet again, the significant improvement in spatial resolution possible with sequential single-molecule localization. In fact, the photoactivated species can be localized in the lysosomal compartments of the live cells with a precision of ca. 39 nm.

## 6. Conclusions

The synthetic versatility of the BODIPY platform permits the installation of photolabile functional groups on the *meso*-position (**1**, **7** and **14**), pyrrole heterocycles (**4**, **17**, **20** and **24**) or boron center (**11**) of the chromophoric platform. The resulting fluorophore–photocage constructs increase their emission intensities or shift their emission bands in response to photoactivation. Such fluorescence changes are a result of the suppression of nonradiative decay pathways (**1**, **7**, **11** and **14**) or modifications in electronic delocalization (**4**, **17**, **20** and **24**) with the photoinduced conversion of reactant into product. Synthetic manipulations also permit the introduction of functional groups on the BODIPY scaffold for the covalent (**7**, **14** and **17**) or noncovalent (**11** and **24**) labeling of intracellular targets. The photoactivatable fluorescence of the resulting intracellular labels can be exploited to capture diffraction-limited images of live cells as well as, when appropriate photochemical and photophysical requirements are satisfied, to reconstruct sub-diffraction images of the labeled intracellular components. The latter imaging protocol requires reiterative sequences of photoactivation, localization and bleaching steps (Figure 2a–i) and is generally facilitated if the fluorescence switching mechanism is based on changes in electronic delocalization. Such photochemical transformations can be engineered to cause significant bathochromic shifts in absorption, enable the selective excitation of the product and switch fluorescence on with infinite contrast in the spectral region where only the latter species emits. In turn, the photoactivation of fluorescence with infinite contrast facilitates the detection of sparse populations of single photoactivated molecules in the presence of a large excess of the starting reactant. Additionally, operating principles based on photoinduced bathochromic shifts also permit the selective bleaching of the localized photoactivated species in the presence of a large excess of the starting reactant. The combination of these photochemical and photophysical properties translates into the ability to localize single molecules with a precision approaching 15 nm in the interior of live cells and visualize organelles with diameters as small as 80 nm. Indeed, BODIPYs with photoactivatable fluorescence are a promising addition to the vast library of existing fluorescent dyes. They may well become the probes of choice for live-cell PALM to enable the routine visualization of subcellular components with nanometer spatial resolution, that is otherwise inaccessible to conventional fluorescence imaging schemes.

## Figures and Tables

**Figure 1 molecules-28-02447-f001:**
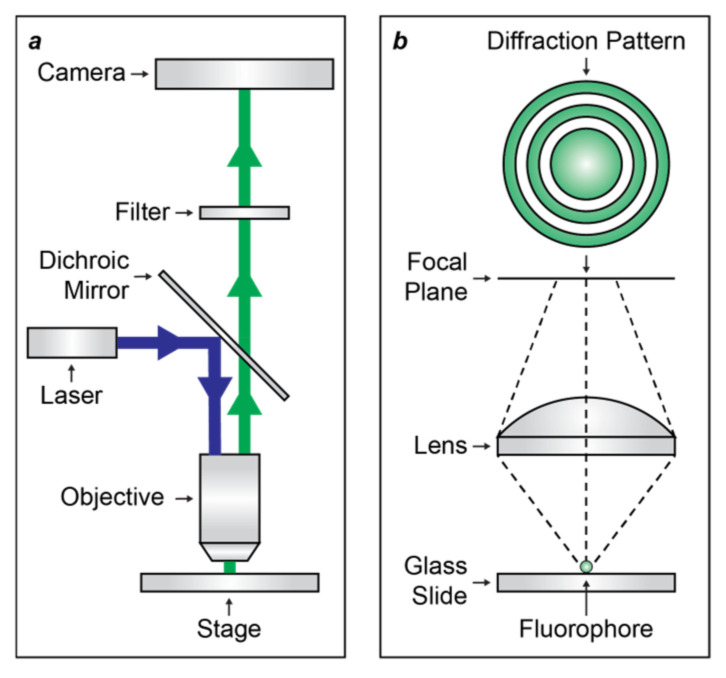
Schematic representations of the main optical components of a conventional fluorescence microscope (**a**) as well as of the Airy pattern (**b**) that the fluorescence of a single fluorophore produces on the focal plane of the objective lens.

**Figure 2 molecules-28-02447-f002:**
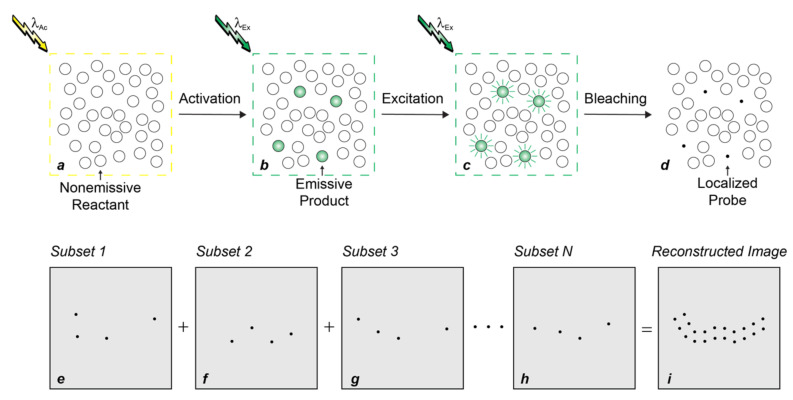
Sequence of steps (**a**–**i**) required for the reconstruction of an image with subdiffraction spatial resolution relying on fluorescence photoactivation and single-molecule localization.

**Figure 3 molecules-28-02447-f003:**
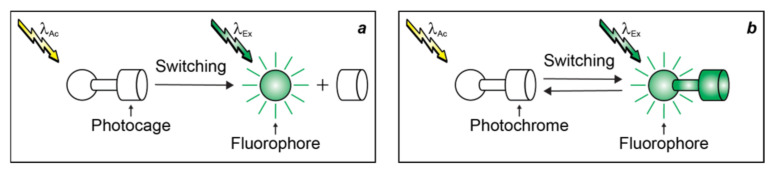
Structural designs for irreversible (**a**) and reversible (**b**) fluorescence photoactivation with fluorophore–photocage and fluorophore–photochrome constructs respectively.

**Figure 4 molecules-28-02447-f004:**
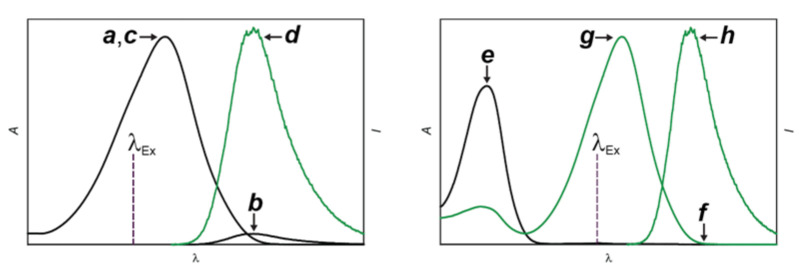
Absorption and emission spectra before (black traces) and after (green traces) the photochemical conversion of a nonemissive reactant into an emissive product expected for fluorescence-photoactivation mechanisms based on intramolecular quenching (**a**–**d**) and bathochromic shift (**e**–**h**) respectively.

**Figure 5 molecules-28-02447-f005:**
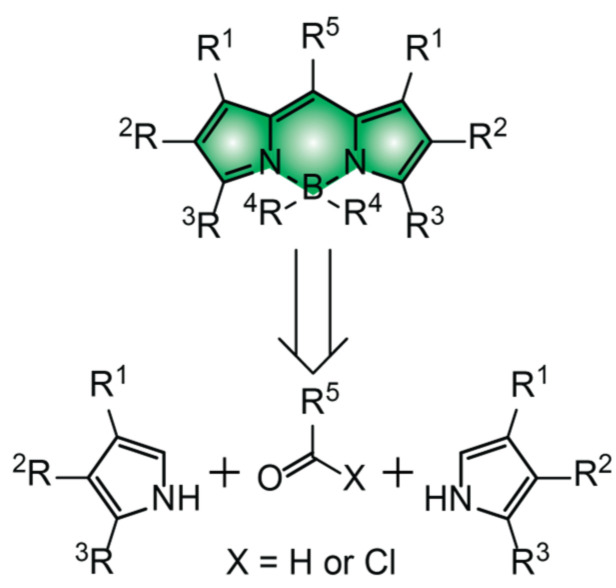
Molecular skeleton of the BODIPY chromophore together with its retrosynthetic analysis.

**Figure 6 molecules-28-02447-f006:**
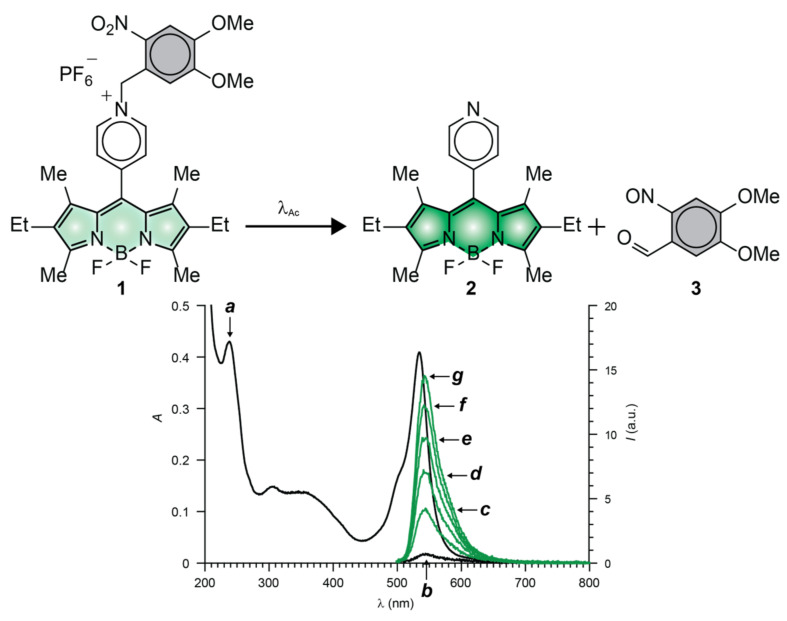
Absorption (**a**) and emission (**b**, λ_Ex_ = 470 nm) spectra (black traces) of an equimolar MeCN solution of **1** and Bu_4_NOH (10 μM) at 25 °C. Emission spectra (λ_Ex_ = 470 nm, green traces) of the same solution after irradiation (λ_Ac_ = 300–410 nm, 3.33 mW cm^™2^) for 5 (**c**), 10 (**d**), 15 (**e**), 20 (**f**) and 25 min (**g**).

**Figure 7 molecules-28-02447-f007:**
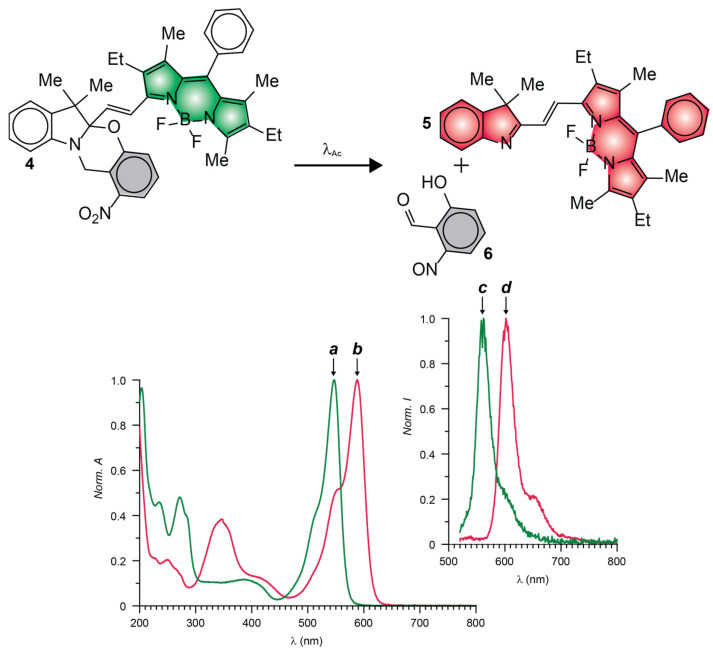
Normalized absorption (**a,b**) and emission ((**c**,**d**), λ_Ex_ = 480 nm) spectra of **4** (green traces) and **5** (red traces) in MeCN at 25 °C.

**Figure 8 molecules-28-02447-f008:**
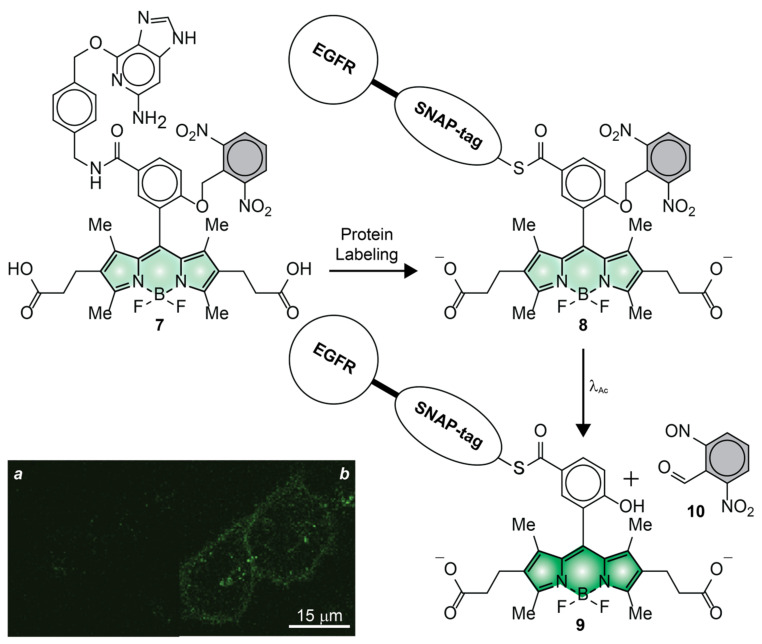
CLSM images of live HeLa cells, expressing EGFR–SNAP-tag labeled with **7**, recorded before (**a**) and after (**b**) photoactivation [adapted with permission from Ref. [54]].

**Figure 9 molecules-28-02447-f009:**
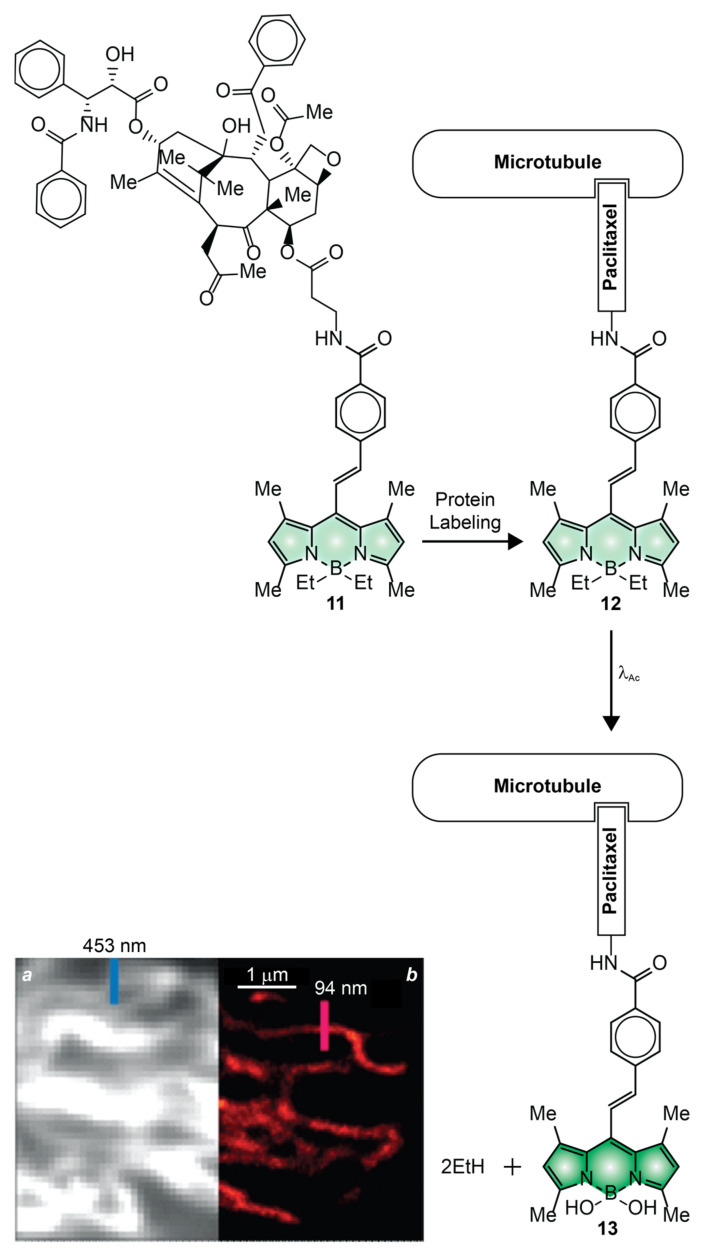
Diffraction-limited (**a**) and PALM (**b**) images of live HeLa cells, stained with **11**, reporting the full width at half maximum of emission-intensity profiles (blue and red bars) measured across the labeled microtubules [adapted with permission from Ref. [50]].

**Figure 10 molecules-28-02447-f010:**
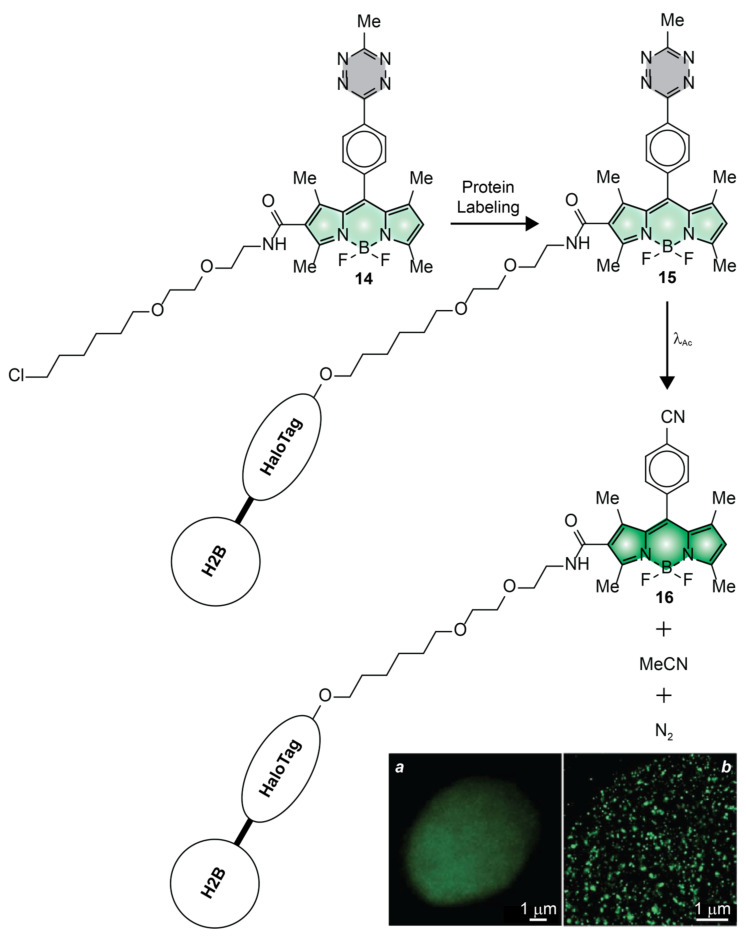
Diffraction-limited (**a**) and PALM (**b**) images of live CHO–K1 cells, expressing H2B–HaloTag labeled with **14** [adapted with permission from Ref. [52]].

**Figure 11 molecules-28-02447-f011:**
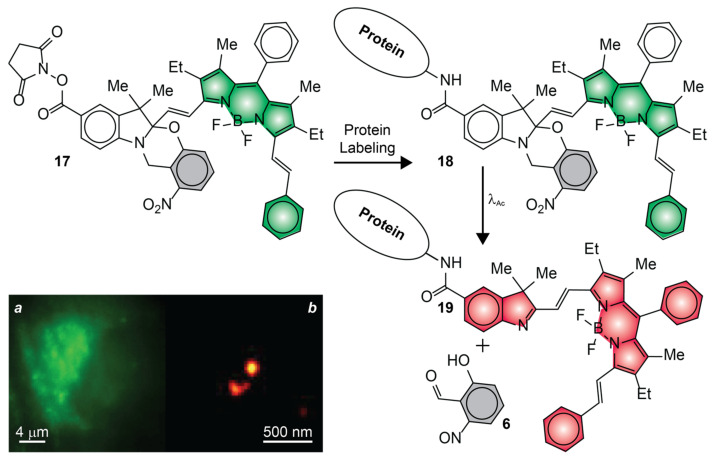
Diffraction-limited (**a**) and PALM (**b**) images of the lysosomes of live COS–7 cells, labeled with **17** [adapted with permission from Ref. [51]].

**Figure 12 molecules-28-02447-f012:**
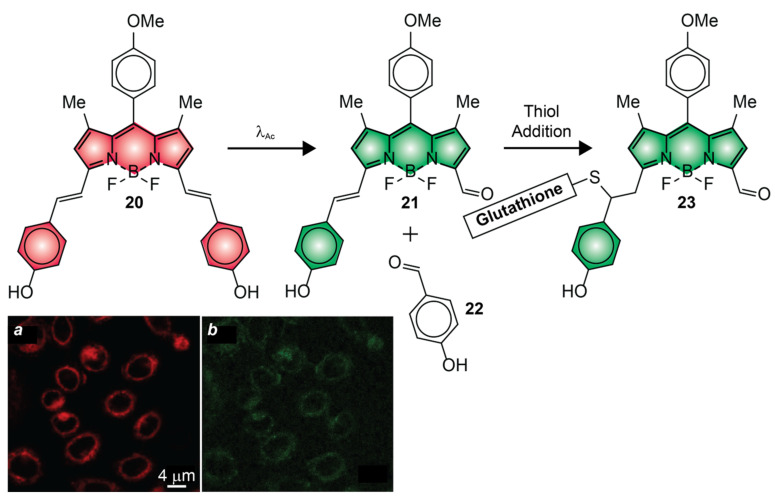
CLSM images of live HeLa cells, labeled with **20**, recorded before (**a**) and after (**b**) photoactivation [adapted with permission from Ref. [64]].

**Figure 13 molecules-28-02447-f013:**
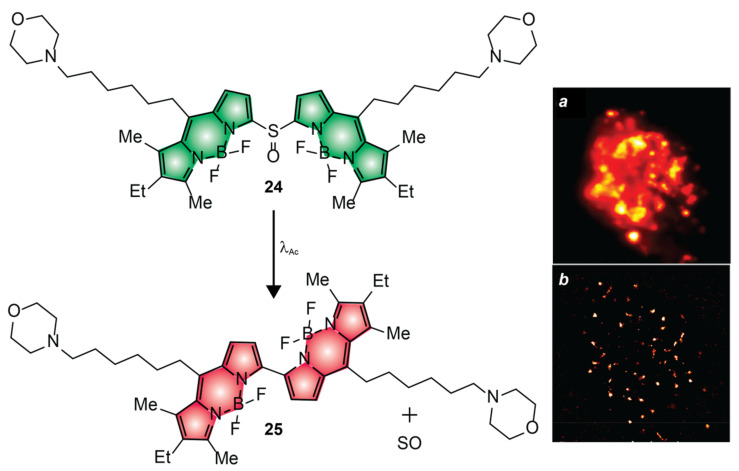
Diffraction-limited (**a**) and PALM (**b**) images of the lysosomes of live RAW264.7 cells labeled with **24** [adapted with permission from Ref. [53]].

**Table 1 molecules-28-02447-t001:** Photophysical parameters.^1^.

	^In^λ_Ab_	^Fi^λ_Ab_	^In^ϕ_Fl_	^In^λ_Em m_	^Fi^λ_Em_	^Fi^ϕ_Fl_	Reference
**1**	534	546	0.005	526	545	0.50	[55]
**4**	548	562	0.07	588	602	0.50	[57]
**7**	526	541	0.001	522	539	0.66	[54]
**11**	498	517	0.002	495	510	0.96	[50]
**14**	491	509	0.002	491	509	0.28	[52]
**17**	608	623	0.89	656	669	0.40	[51]
**20**	654	673	0.90	590	630	— ^2^	[64]
**24**	514	561	0.20	630	701	0.23	[53]

^1^ Wavelengths at the absorption maxima of the initial (^In^λ_Ab_) and final (^Fi^λ_Ab_) states; wavelengths at the emission maxima of the initial (^In^λ_Em_) and final (^Fi^λ_Em_) states; fluorescence quantum yields of the initial (^In^ϕ_Fl_) and final (^Fi^ϕ_Fl_) states. Measured in MeCN, containing equimolar amounts of Bu_4_NOH, for **1**, MeCN for **4** and **14**, PBS for an analog of **7**, lacking the SNAP-tag ligand, MeOH for **11**, THF for **17**, MeOH/H_2_O (1:1, *v*/*v*) for **20** and CH_2_Cl_2_ for **24**. ^2^ Not reported.

## Data Availability

Not applicable.

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
