# Peer review of "Photoactivatable BODIPYs for Live-Cell PALM"

_molecules, 2023, doi:10.3390/molecules28062447_

Round 1

Reviewer 1 Report

The authors overview the state or art of a nanospectroscopy technique based on photoactivatable probes where BODIPY is used as fluorophore. They start describing the technique and its suitability to overcome the diffraction limit, and after that, they detail the photophysical and photochemical requirements of the probes and the available detection mechanism based on suppression of non-radiative processes or induction of bathochromic shifts of the fluorophore after removing the photocage upon irradiation with suitable wavelength. Finally, they describe the representative examples reported in the bibliography of labelled BODIPY-based photoactivatable probes, via at boron or meso position modifications, and their performance under conventional fluorescence microscopy or to construct sub-diffraction fluorescent images.

I strongly recommend the acceptance of the manuscript. It is very well written and organized and the scope is a topical subject with huge future perspectives to explore. Just a couple of comments or doubts

I think that the sentence before Figure 4 in page 4 (lines129-131) needs to be better explained for the sake of clarity. Before such point, the photoactivation of the PAFs is described in terms of suppression of non-radiative deactivation, and in such sentence is suddenly introduced an alternative mechanism based on the induction of bathochromic shifts. It is true that later on in section 3 is well described, but I think that a brief explanation shoul be added when it is first mentioned in section 1

In section 3, line 148 the molar absorption coefficient is given in mM-1 cm-1. I think that the factor m is not correct here, because it accounts as a factor of 10-3, whereas the real value is 70000 M-1 cm-1.

Reviewer 2 Report

I have thoroughly reviewed Y. Zang et al.'s manuscript, which is a mini-review on the potential of using photoactivated BODIPYs for subdiffraction imaging of cellular substructures in living cells based on PALM. While I believe the manuscript will be of interest to the journal's readers, I cannot recommend it for publication without significant improvements by the authors.

Key comments:

1. The authors did not consider what classes of phosphor other than BODIPY are used for Live-Cell PALM .
2. What other methods of fluorescence quenching are there besides introducing quencher molecules into the BODIPY core?
3. There is no systematization of structural features and necessary physicochemical properties of compounds, which would be of interest for Live-Cell PALM, which is typical for Reviews.

And finally, p. 4 - check the dimension of the molar absorption coefficient and figure 13 should be carefully checked.

Reviewer 3 Report

The manuscript entitled "Photoactivatable BODIPYs for Live-Cell PALM" reviews the imaging application of photoactivatable fluorophores based on BODIPY. Factors including design principles and electronic effects have been highlighted. However, the overall subject is interesting; the current review has some flaws and lacks some critical details that need to be addressed carefully and thoroughly before the paper can be endorsed for publication.

1. Before section 1, authors are suggested to include an introduction section discussing state-of-the-art and highlighting the novelties of this review with respect to other published reviews.

2.     Only seven examples have been discussed in this review. Authors should include more examples to highlight this research area's utility.

3.     All figures are of very low quality; they should be enhanced.

4.     Figure 5: Substitients R1 to R5 should be defined.

5.     Abbreviations used in the manuscript should be defined.

Round 2

Reviewer 2 Report

The authors addressed most of my questions, but some comments were not adequately addressed. Specifically, while the authors presented the spectral properties of the compounds in Table 1, a systematization of properties and structural features is missing to help readers understand whether the compounds they synthesized are suitable for Live-Cell PALM.

I am also confused by the presence of sulfur monoxide as a product of the reaction shown in Figure 13. I could not find any information about this in the original article [10.1021/jacs.2c08947].

Reviewer 3 Report

The authors have revised their MS satisfactorily.
